# Quantification of the Effects of Electrical and CO_2_ Stunning on Selected Quality Attributes of Fresh Pork: A Meta-Analysis

**DOI:** 10.3390/ani12141811

**Published:** 2022-07-15

**Authors:** Andrzej Zybert

**Affiliations:** Faculty of Agrobioengineering and Animal Husbandry, Institute of Animal Science and Fisheries, Siedlce University of Natural Sciences and Humanities, Prusa 14, 08-110 Siedlce, Poland; andrzej.zybert@uph.edu.pl

**Keywords:** pigs, stunning, meat quality, meta-analysis

## Abstract

**Simple Summary:**

Stunning is an important and statutory pre-slaughter process that aims to induce a state of unconsciousness and insensibility to pain just before sticking and bleeding out. In Europe, two main stunning methods that are applied at pig slaughter are electrical (ES) and gas stunning with high-concentration carbon dioxide (CO_2_). Stunning is not only crucial in terms of the welfare of pigs at slaughter, but it may also have effects on meat quality. Various studies have compared these two methods with respect to meat-quality traits. In several studies, stunning with CO_2_ has been shown to improve meat quality in comparison with ES; however, other research indicated that ES enhanced the meat quality as opposed to CO_2_ or had no significant effect. The current study analysed the combined results of independent research and experiments to achieve quantitative evidence of the effects of two stunning methods, electrical and gas with a high concentration of carbon dioxide, on selected quality attributes of fresh pork. The results of the meta-analysis revealed that relative to CO_2_ stunning, overall, ES pigs had a lower pH_1_, higher drip loss (DL) and lightness (L*). However, the greater alterations in these traits were achieved with the application of the head-to-back (HB) electrical-stunning method and conventional chilling (Conv.) of the carcasses. The results of the meta-analysis provide evidence that differences between these two stunning methods in DL and L* may be diminished by the application of the head-only (HO) or head-to-body (HBO) method, followed by the fast chilling of the carcasses.

**Abstract:**

Stunning is a statutory pre-slaughter process that may affect the quality of pork. The objective of this study was quantification of the effects of stunning (ES vs. CO_2_ stunning) on selected quality attributes of pork, using a meta-analytical approach. Data from 18 publications with 46 individual experiments were combined using a random-effect model to estimate the effect size of stunning on the initial and ultimate pH (pH_1_, pH_u_); drip loss (DL); colour (lightness—L*, redness—a*, yellowness—b*); and tenderness (expressed as Warner–Bratzler shear force, WBSF) of pork. In overall, loins from ES showed significantly lower pH_1_ (by 0.08 units); greater DL (by 0.68 p.p.); higher L* (by 1.29 units); and a* (by 0.80 units) as they compared to those from CO_2_ stunning. In subgroups, a greater-than-overall negative change in pH_1_, pH_u_, DL, L* and a* was detected with the application of the head-to-back (HB) stunning method. Additionally, alterations in DL and L* may be magnified with the application of conventional chilling (Conv.) to ES pigs. There was no effect of stunning on WBSF but, due to a low number of research in the database, the reliability of these results may be misleading. These results provide evidence that the differences between these two stunning methods in DL and L* may be diminished by the application of the head-only (HO) or head-to-body (HBO) method, followed by the fast chilling of carcasses.

## 1. Introduction

According to European Council Regulation No 1099/2009, all farm animals must be stunned prior to slaughter. Stunning is a process that aims to induce unconsciousness and insensitivity to pain just before sticking and bleeding out. In Europe, three stunning methods are allowed for pigs: mechanical, electrical and gas. Among these methods, two of them, electrical and gas stunning with a high concentration of carbon dioxide (CO_2_), are commonly used by meat processors [1]. The principle of electrical stunning (ES) is to pass an electrical current with sufficient strength and duration through the brain to depolarize the neuronal membranes. This leads to an epileptiform seizure and loss of consciousness [2,3]. According to EU regulation 1099/2009, pigs must be stunned with a minimum current of 1.3 amperes, maintained for at least 3 s, and with a voltage of at least 240V. In the gas-stunning method, pigs are exposed to high (at least 80%) concentrations of CO_2_ in air. A high concentration of CO_2_ leads to hypercapnic hypoxia, reduced blood pH levels and acidification of the cerebrospinal fluid. This results in the acidification of brain cells and depression of brain activity, causing loss of consciousness [4]. Stunning, especially from the perspective of EU Regulation 1099/2009 requirements, is crucial to the welfare of pigs. However, stunning may also affect the quality of meat.

In Europe, where pork consumption is relatively high, the quality of meat is an important aspect that is linked with the preferences of pork consumers, their acceptance of meat, and their intention to purchase [5]. Furthermore, quality-oriented consumers are willing to pay more for high-quality pork [6]. The quality of fresh pork is a very complex issue and difficult to define [7,8,9]; overall, it covers a variety of properties that are decisive for the suitability of the meat for use as food or cooking. The quality is also defined as a set of cues that are perceivable by consumers when purchasing and consuming meat [7,10,11]. At the moment of purchase, the most important cues are such attributes as the amount of fat, colour, and water-holding capacity, while tenderness determines quality at the moment of consumption [9]. These quality attributes vary with the breed, production system, nutrition and preslaughter handling; however, a slaughter procedure with stunning may also change the quality of fresh pork [7,8]. Thus, the application of technologies enabling the production of meat with low variability in traits is still an important issue for the meat industry.

The main advantage of the ES method is that it is quick to apply. Furthermore, it ensures an immediate loss of consciousness, while in CO_2_ stunning, unconsciousness occurs gradually (on average, after 60 s of exposure to 90% CO_2_ [12]). However, the application of electrical stunning can lead to a higher incidence of haemorrhages (petechiae and ecchymoses) and bone fractures in the forelegs and vertebral column, in comparison to CO_2_ stunning [13,14,15]. Furthermore, electrical stunning may result in a lower pH at early post-mortem, greater drip loss, and a lighter colour [13,14,16,17]. On the other hand, several authors found no conclusive evidence of the stunning impact on drip loss and the colour of pork. Hambrecht et al. [18] showed that loins from CO_2_-stunned pigs had significantly greater drip loss and were lighter than those from electrical stunning. Others [15,19,20] found no effect of stunning method (electrical vs. CO_2_) on drip loss and the lightness of pork. Thus, the results from these individual studies vary both in magnitude and the direction of the stunning method effect. In consequence, the qualitative assessment of stunning effect on meat-quality traits is very difficult. A meta-analysis is a statistical procedure that combines and summarizes the results of independent research into a single, overall measure of the effect [21]. In the past, several researchers utilised meta-analysis to examine the influence of various pre- and post-slaughter factors on meat quality [22,23,24,25,26]. However, there is no published meta-analysis on the effect of stunning on the quality of pork. Thus, the objective of this study was to apply a meta-analysis to quantify the effects of two stunning methods, electrical and gas with high concentration of carbon dioxide, on selected quality attributes of fresh pork.

## 2. Materials and Methods

### 2.1. Literature Search and Selection Criteria

A literature search was conducted via Scopus, Google Scholar, EBSCO, ProQuest, and Science Direct digital databases. Backward search referred to the manual searches of reference lists from papers, review articles, and conference proceedings, in order to identify the studies that provided data on the meat quality of pigs that were submitted to electrical and high CO_2_ stunning. The literature search focused only on these studies, in which the experimental design included a comparison of electrical with high CO_2_ stunning. Studies were included into the database if they met the following criteria: (1) published in English or with an English abstract; (2) carried out on pigs; (3) evaluated meat-quality traits; (4) quality traits were measured in longissimus muscle; (5) covered the description of meat-quality measurements; (6) provided data that were sufficient for determining the effect size of treatment outcomes as the number of animals, mean values, and the measure of variance expressed as standard deviation (SD) or standard error (SE) for extracted outcomes. When the effect of stunning was explored within a study in separate experiments, then these experiments with relevant outcomes were considered as separate studies. The initial search gave 22 studies. Among these references, some provided only an average estimate of the outcomes for each stunning group, but without the sample size in groups or the measures of variance [27,28,29,30]. Finally, the screening process resulted in the selection of 18 publications with 46 individual experiments (Table 1).

### 2.2. Data Base Preparation

The database included the author name; year of publication; number of animals; where possible, the type/method of ES application (head-only, HO; head-to-back, HB; and head-to-body, HBO); parameters of the stunning process; chilling method (conventional, with temperatures of 1–4 °C with air velocity below 1 m/s, and fast/accelerated chilling, with pre-chilling phase in temperatures from −10 °C to −35 °C and air velocity of 3–5 m/s); and mean with corresponding variability measures of quality outcomes of each treatment group. When the study provided a pooled SD or SE estimate, that estimate was used for both treatment (stunning-related) groups. The following quality outcomes were extracted from the studies: initial pH (pH_1_) measured between 45 min and 60 min after the slaughter; ultimate pH (pH_u_—measured 24 or 48 h after the slaughter); meat colour determined in CIE colour system and expressed as lightness (L*), redness (a*), and yellowness (b*); drip loss expressed as a percentage of weight loss after 24 h or 48 h of storage at 4 °C relative to the initial weight of a muscle sample; and Warner–Bratzler shear force (WBSF, kg). When the WBSF estimates were displayed within a study in Newtons (N), these averages and relevant measures of variance were converted into the kilograms force. The descriptive statistics of the data that were included in the meta-analysis references has been presented in Table 2.

### 2.3. Statistical Analysis

All the statistical analyses were completed with PQ-Stat 1.6.4.188 statistical software (PQStat Software, Poznań, Poland). The effect size, reflecting the magnitude and the direction of the treatment effect, was calculated as the weighted mean difference (WMD). The WMD is the mean difference between two groups that are weighted by their sample size [41]. The positive effect size indicated that the quality outcomes were greater in the ES group, whereas a negative effect size indicated that the quality outcomes were higher in the CO_2_ stunning group. In meta-analyses, studies are usually combined using fixed and random-effect models [21,42]. A fixed model assumes a common treatment effect among combined studies. However, the results of multiple studies usually vary due to differences in animals/breeds, experimental design, treatment parameters, or other unknown factors. Thus, a random-effect model, which allows variability (heterogeneity) among studies, was adopted to estimate the overall effect size, 95% confidential interval (95% CI), and statistical significance of the effect [21,43,44]. Heterogeneity, reflecting how much the responses to the treatment differ across studies, was determined using I^2^ statistics. I^2^ ranges from 0% to 100% and measures the proportion of inconsistency that cannot be explained by chance alone. I^2^ values of 25%, 50%, and 75% are considered as small, moderate, and high heterogeneity [45]. Publication bias was examined using Egger’s test [21]. The presence of homogeneity between studies or the identification of sources of heterogeneity improves the understanding of responses to treatment and interpretation of results [46]. The presence of heterogeneity across studies is a major concern in a meta-analysis. In the presence of homogeneity, where there is more assurance that in future experiments, the treatment has a similar effect, but in the existence of unexplained heterogeneity, the effect of the treatment in future experiments is harder to predict [47]. The common method that is used to explore the sources of heterogeneity in a meta-analysis is the subgroup analysis [46,48]. Therefore, the data were split into subgroups that were diversified by the type/method of ES that was associated with the placement of electrodes (head-only, HO; head-to-back, HB and head-to-body, HBO) and chilling method (conventional and fast/accelerated), and each subgroup was subjected to a separate meta-analysis. Nevertheless, in the presence of unexplained heterogeneity between studies, Higgins et al. [44] recommended reporting the prediction interval (PI), which provides a more complete summary of a random-effect meta-analysis. PI estimates the range of effect estimates for 95% of similar studies. Within variability across studies, PI covers a wider range then 95% CI. Additionally, it provides a more informative picture of the treatment effect in comparison to findings that are focused on 95% CI [49].

## 3. Results

The meta-analysis of all available research detected that, overall, stunning had a significant effect on the pH_1_, drip loss, lightness, and redness of fresh pork (Table 3).

Overall, the meta-analysis indicated that pigs that were submitted to ES had a small in magnitude but significantly (*p* ≤ 0.05) lower pH_1_ (by 0.08 units) in comparison to pigs that were submitted to CO_2_ stunning (Table 3). There was no publication bias (*p* > 0.05). However, the I^2^ statistics indicated the presence of high heterogeneity (87.33%) between studies. The sub-group meta-analysis indicated that the method of electrical stunning altered the effects of ES in comparison with CO_2_ stunning. However, a lower than overall pH_1_ was detected only when HB-stunned pigs were compared with pigs from CO_2_ stunning. Two other types/methods of ES had no effect on pH_45_ (Table 3); however, there was substantial heterogeneity between studies, and so PI was also reported. PI that was computed for pH_1_ was wider then an overall 95% CI and ranged from −0.418 to 0.254.

There was no significant effect of stunning on the overall ultimate pH, although there was high heterogeneity between studies (I^2^ = 93.5%) and publication bias (*p* < 0.01). In subgroups, only the type/method of electrical stunning had an effect on ultimate pH. Nonetheless, a significantly (*p* ≤ 0.05) lower than overall pH_u_ (0.05units) was only seen in HB-stunned pigs, as compared with pigs from CO_2_ stunning. However, there was high inconsistency between studies. Two other types of ES and chilling methods had no effect on ultimate pH.

Overall, the meta-analysis indicated that loins from ES pigs had significantly (*p* ≤ 0.01) higher DL (in average 0.68 p.p.) in comparison to pigs that were submitted to CO_2_-stunning. (Table 3). The I^2^ statistics indicated that there was substantial heterogeneity across all studies, as well as publication bias, which indicated the overestimation of the effect size. In subgroups, the meta-analysis indicated that both HO and HB methods altered the effects of ES in comparison with CO_2_. However, the greatest increase in DL (1.82 p.p.) was achieved with the application of HB stunning. When the data were split into the two chilling-related subgroups, a significant increase in DL of 0.94 p.p. was achieved with the application of conventional chilling. Fast chilling had no effect on DL (Table 3). Additionally, all subgroups held moderate or higher I^2^ estimates, which implied that other factors may have been responsible for the variability across studies; therefore, PI was also reported. For DL, PI ranged from −0.722 to 2.076.

Overall, the meta-analysis showed that loins from pigs that were electrically stunned had significantly higher L* (1.29 units) and a* relative to the animals that were stunned with CO_2_ (Table 3). However, there was high inconsistency between studies, but without publication bias (*p* > 0.05). Thus, other factors may have impacted variability in the colour of pork loins. In subgroups, the meta-analysis indicated that the method of electrical stunning altered the effects of ES in comparison with CO_2_ stunning. However, the overall trend of an increase in L* was confirmed (*p* ≤ 0.01) only in the HB group. Additionally, there was high homogeneity across studies (Table 3). Two other methods hold high heterogeneity without the effects on L*. The chilling method also had an effect on L*, but the trend of an increase in L* was detected only when fast chilling was applied to ES pigs. In subgroups, only the HB method had no effect on a*. Two other ES methods and both chilling methods modified the effect of ES in comparison with CO_2_ stunning. However, in most comparison groups, an increase in a* with ES application was similar to an overall effect size. Stunning had no effect on the tenderness of pork loins; however, only 15 experiments from five studies were included in the meta-analysis. Thus, the reliability of these results may be misleading.

## 4. Discussion

Meat quality is a complex of traits that together are responsible for consumer satisfaction and willingness to buy [8]. For fresh meat, intended for consumption, the most important are those that are associated with water holding capacity, colour, and tenderness [9]. A key role in the variability of these traits is the rate and extent of pH-decline post-mortem. [50]. Normally, in longissimus muscle, pH declines gradually from 7.2 in living muscles to 5.5–5.6 at 24 h post-mortem. Muscles with hastened pH decline exhibit a pH of less than 5.8 within the first hour after slaughter. This alteration in pH decline adversely influences water-holding capacity, meat colour, and tenderness [51,52]. Several studies [14,15,17,34,35,38] showed that pigs that were submitted to ES had lower pH_1_ in comparison to those that were submitted to CO2 stunning. Others [18,40] reported a higher pH after ES in comparison with CO_2_-stunning. The results of this meta-analysis indicated that ES pigs had an overall significantly lower pH_1_ than pigs that were submitted to CO_2_ stunning. However, there was high heterogeneity across all studies. There are several reasons why ES may accelerate pH decline post-mortem, resulting in lower—in comparison with CO_2_ stunning—pH_1_. Firstly, electrical stunning is a more stressful method in comparison to CO_2_ stunning [28,33,36]. Additionally, electrical stunning requires restraining, which is also stressful for pigs [36,53]. Physical stress just before or during stunning accelerates pH decline early post-mortem due to an elevated release of catecholamines into the blood and increased muscle activity/contraction [54]. This may also result in pale, soft, exudative (PSE) meat. The formation of PSE meat is a consequence of a genetic mutation in calcium-regulating protein, ryanodine (RYR1), also known as halothane gene [55,56]. It is well known that the RYR1 sensitivity allele (T) has a negative impact on pH decline early post-mortem and the prevalence of PSE meat [57]. Channon et al. [14,35], Gispert et al. [31] and Velarde et al. [13,32] found that electrical stunning produced more PSE meat compared with CO_2_ stunning. However, the frequency of PSE meat increased when electrical stunning was applied to stress-sensitive animals [14,31,32]. Channon et al. [14] also found that differences between CO_2_ and electrical stunning disappeared in stress-resistant animals.

Nonetheless, other factors may influence the pH_1_ of pork loins. A potential source of variability in pH decline post-mortem may be the method of electrical stunning. There are three types/methods of electrical stunning, head-only (HO), head-to-back (HB) and head-to-body (HBO) which use two or three electrodes. In the HO method, the electrodes are applied to each side of the head, between the eye and the ear, or just below ears. In the HB method, the front electrode is placed between ears while the rear electrode is applied to the chest, behind the position of the heart or on the back [3,58]. An automatic HBO method uses three electrodes. A pair of electrodes is automatically placed just below the ears, while a third, heart electrode, is applied 0.7 s after the head electrodes, behind the left shoulder [18]. The application of the heart electrode leads to cardiac arrest and a reduction in clonic convulsions, due to the inhibition of spinal nerve function [59]. For pH_1_, the sub-group meta-analysis indicated that the method of electrical stunning modified the effects of ES in comparison with CO_2_ stunning. However, a lower than overall pH_1_ was only seen in HB-stunned pigs, as compared with those from CO_2_ stunning. The results of individual studies examining the effects of different stunning methods/types on pH decline post-mortem are not conclusive. Channon et al. [34] found that HB-stunned pigs had a lower pH at 40 min after slaughter than those from HO stunning. In contrast, van de Perre et al. [60] showed that pigs that were stunned with the method using three electrodes revealed significantly lower pH at 30 min after slaughter, in comparison to those that were stunned with HO and HB methods using two electrodes. As concluded, this was probably due to the higher noise level that was recorded in the group in which the three-electrode method was used.

Other than the rate of pH change, the extent of pH decline post-mortem and the level of ultimate pH play a key role in the development of water-holding capacity, colour, and tenderness. The extent of pH decline is largely determined by the amount of glycogen in muscle at slaughter [61,62]. After slaughter, during anaerobic glycolysis, glycogen is converted to lactate and H^+^, which accumulate and decline the pH of meat. The cessation of post-mortem glycolysis occurs with the depletion of glycogen in muscles or the inactivation of enzymes controlling glycolysis [50]. Several studies [14,15,17,20,34,35] showed no differences in pH_u_ when pigs that were submitted to ES were compared with pigs that were submitted to CO_2_ stunning. Furthermore, Channon et al. [14] reported no effect of stunning, both on the rate of pH decline from 40 min to 24 h post-mortem and lactate and glycogen concentration at 40 min to 24 h after slaughter. However, the relative rate of pH decline to 24 h post-mortem may vary with the method/type of ES. Channon et al. [34] found that HB-stunned pigs had a significantly higher relative rate of pH decline (from 40 min to 24 h post-mortem) than HO-stunned pigs. However, these findings differ with Channon et al. [35] who found, that HO stunned pigs had a higher relative rate of pH decline, compared with HB and CO_2_-stunned pigs. The results of this meta-analysis showed no effect of stunning on the overall pH_u_. However, the sub-group meta-analysis detected that the method of electrical stunning modified the effects of ES in comparison with CO_2_ stunning. Nonetheless, a lower than overall pH_u_ hadonly HB-stunned pigs, as they compared with those from CO_2_ stunning. The ultimate pH may also vary with the chilling condition, which can affect the activity of the enzymes regulating post-mortem glycogenolysis and buffering capacity of muscles [63,64] In Europe, the most common cooling medium that is used in the chilling of pork sides is air [65]. The methods of chilling vary with the temperature and speed with which the cooling medium circulates over the warm carcasses. Conventional chilling uses temperatures of 1–4 °C with an air velocity below 1 m/s, while fast/accelerated chilling, in the pre-chilling phase, uses temperatures from −10 °C to −35 °C with an air velocity of 3–5 m/s [65,66]. The increase in heat removal from carcasses, coupled with temperature decline in muscles, prevents the formation of inherent pork due to a decrease in the rate and extent of post-mortem metabolism and pH decline [64,67,68,69,70]. The accelerated chilling, in comparison to conventional chilling, has an ability to decrease the relative rate of pH decline and increase ultimate pH [26,69,70]. Nonetheless, the sub-group meta-analysis indicated that the chilling method had no effect on variability in pH_u_ between ES and CO_2_-stunned pigs.

The drip loss is affected by numerous factors including genetic background, feeding, preslaughter handling, stunning, and chilling method [71,72]. All of these factors affect different biochemical and physical processes that occur during the conversion of muscle to meat, such as the rate and extent of pH decline during slaughter, the temperature of muscles immediately after slaughter, and over the time post-mortem; the shrinkage of the myofibrils and sarcomere length, which changes interfilamentous spacing; the development of extracellular spaces and channels; and the increase in cell membrane permeability, which are believed to influence the distribution of water within the muscle [72,73]. Several studies showed that pigs that were submitted to electrical stunning had significantly higher drip loss than those from CO_2_ stunning [14,17,34,35]. In contrast, Hambrecht et al. [18] and Terlouw et al. [40] found that pigs that were exposed to CO_2_ stunning had greater drip loss than those that were exposed to electrical stunning. However, Bertoloni et al. [19] and Marcon et al. [15] reported no effect of stunning on drip loss. The results of this meta-analysis indicated that loins from pigs that were electrically stunned had on average a significantly higher (0.68 p.p.) DL relative to the animals that were CO_2_ stunned. The effect of electrical stunning on drip loss is associated with early post-mortem metabolism and higher levels of phosphocreatine determining initial pH drop, rather than the rate of pH decline [33,74], even in populations of stress-resistant pigs [36,74]. Additionally, electrical stunning that is coupled with spastic muscular contraction and faster rigor-mortis development due to more rapid pH decline may also result in an increase in water distribution in meat [17,75]. Finally, electrical stunning increases the disruption of cell membranes and the redistribution of extracellular water, resulting in greater drip loss [76] An advantage of high-CO_2_ stunning is handling pigs in small groups, and the possibility of stunning them without restraint. However, CO_2_ stunning requires a longer period to induce unconsciousness. An induction period is very stressful for pigs [4,77]. This may lead to acceleration of muscle metabolism early post-mortem and then an increase in drip loss [40]. Drip loss may also vary with the method/type of ES, although the results of individual research are not fully conclusive. Channon et al. [34] showed that HB-stunned pigs had a significantly greater drip loss than those from CO_2_ stunning; however, no differences in drip loss were found between HO and CO_2_-stunned pigs. Previously, Channon et al. [14] found that HO-stunned pigs had greater drip loss than those that were submitted to electrical stunning. In the present study, the sub-group meta-analysis detected that both HB and HO stunning had an effect on drip loss. However, the greatest increase in DL (twice as large as overall) was computed for the HB sub-group. This subgroup also held medium heterogeneity. Some studies have shown that accelerated air chilling has the potential to decrease drip loss [26,69,78,79,80]. The chilling-dependent mechanism controlling water distribution and mobility in muscles is associated with the effect of temperature on post-mortem metabolism and temperature-induced structural changes [63]. This mechanism partially explains the computed results of the sub-group meta-analysis. In this research, the sub-group meta-analysis indicated that the overall trend of an increase in DL with the application of ES may have been magnified by the use of conventional chilling.

The colour of fresh meat is determined by the concentration and the chemical form of myoglobin, whereas the lightness is determined by the structural attributes. Several studies have demonstrated that the application of electrical stunning may result in a lighter [3,15,17,34,37], more red [17,20,37], and yellow [17,20] pork colour, in comparison with CO_2_ stunning. In contrast, Hambrecht et al. [18] reported that the application of CO_2_ stunning may result in lighter pork in comparison with ES. Others found no differences in lightness [14,19], redness, and yellowness [15,32,34] when these two methods were compared. In the present study, the meta-analysis showed that loins from pigs that were electrically stunned had on average a significantly higher (1.29 units) L*and a* relative to the animals that were stunned with CO_2_. According to Zhu and Brewer [81], such a high increase in lightness is detectable by the consumer. The effect of ES on pork colour is associated with the pH/temperature history of muscles post-mortem. A rapid pH decline early post-mortem, when the temperature of muscles is still high, may result in the greater denaturation of proteins. Such denaturation results both in elevated water loss and altered light scattering [7,73]. Additionally, low pH and high temperature accelerates the inactivation of oxygen-consuming enzymes and induces the oxygenation of myoglobin to oxymyoglobin [37]. Muscles with normal a rate of pH decline post-mortem but exhibiting an extended glycogenolysis may achieve a low ultimate pH, approaching an isoelectric point of the major muscle proteins. This promotes the reduction in space within the myofibril and relocation of water, and in turn, affects the light scattering properties of the meat [73,82]. Additionally, the sub-group stratification analyses showed that the trend of an increase in L* with the application of ES was magnified only with the application of the HB method. High homogeneity in this group indicates reliability of the result. Two others ES methods showed an increase in a* within a range that was close to the overall effect. Thus, these results, in association with those that were detected for pH_u_ and drip loss, may suggest that the mechanism of pork discoloration that is induced by the HB-stunning method is probably partly associated with a more rapid pH decline post-mortem, promoting structural changes and water flow from muscles, which in consequence increases the lightness of meat. The colour of pork may also vary with the chilling method. Fast chilling produces darker [80,83], less red [83,84], and yellow [83,85,86] pork in comparison to conventional chilling. According to Lindahl et al. [87], these colour differences are associated with a faster decline in temperature in fast-chilled sides, which slows pH fall and preserves against denaturation oxygen-consuming enzymes. This mechanism partially explains the computed results of the sub-group meta-analysis. In this research, L* was increased (*p ≤* 0.05) by ES relative to CO_2_ stunning only within conventional chilling, but it was not influenced (*p* > 0.05) by the fast chilling.

Meat tenderness is a complex attribute, determined by a variety of factors. Among them, the concentration of connective tissue, integrity/degradation of cytoskeletal proteins, and sarcomere length are the main factors that are responsible for variability in meat tenderness [88]. The temperature of muscles and the rate and range of pH decline also affect meat tenderness through the effect on sarcomere length and degradation of cytoskeletal proteins [89]. The rapid pH decline early post-mortem, when the temperature of muscles is still high, can denature myofibrillar and sarcoplasmic proteins and proteolytic enzymes, and result in a soft texture [14]. In turn, the slow pH decline with a rapid temperature decline that occurs before rigor-mortis onset may contribute to sarcomere shortening. This condition of decreasing meat tenderness is known as cold shortening. Channon et al. [35] and Marcon et al. [15] found no differences in tenderness when electrically stunned pigs were compared to those from CO_2_ stunning. Rees et al. [17] reported that loins from electrically stunned pigs were more tender than those from CO_2_-stunned pigs. However, Rees et al. [17] and Marcon et al. [15], found no differences in sarcomere length, myofibrillar fragmentation, or protein denaturation due to stunning method. In the present meta-analysis, stunning had no effect on WBSF; however, the interpretation of this finding should be careful, due to a low number of studies in the database.

## 5. Conclusions

This meta-analysis, combining the results of 18 publications with 46 individual experiments, indicated that relative to CO_2_ stunning, the loins of ES pigs had an overall lower pH_1_, greater DL, and higher L* and a*, but due to the presence of unexplained heterogeneity, these were within the range of computed PI. Furthermore, an overall trend of decreasing pH_1_ and increasing DL and L* was magnified with the application of the HB-stunning method. Moreover, DL and L* were increased when conventional chilling was applied to ES pigs. The results of the meta-analysis revealed evidence that differences between these two stunning methods in DL and L* may be diminished by the application of the HO or HBO method, followed by the fast chilling of carcasses.

## Figures and Tables

**Table 1 animals-12-01811-t001:** Summary of the studies included in the meta-analysis.

Reference	Exp.	Number of Animals		ES Type	Chilling	Variables
		ES	CO_2_			
Channon et al. [17]	1	40	40	HO	Conv.	pH_1_
	2	40	40	HO	Conv.	
Channon et al. [14]	1	9	9	HO	Conv.	pH_1_, pH_u_, DL, L*, WBSF
	2	10	9	HO	Conv.	
	3	10	9	HO	Conv.	
	4	9	9	HO	Conv.	
Gispert et al. [31]	1	427	731	HO		pH_u_
	2	729	731	HO		
	3	629	731	HO		
	4	559	731	HO		
Velarde et al. [13]		1183	1212	HBO	Conv.	pH_u_, L*, a*, b*
Velarde et al. [32]		135	178	HBO	Fast	pH_u_, L*, a*, b*
Bertram et al. [33]		2	2	HO		pH1, DL
Channon et al. [34]	1	10	10	HB	Conv.	pH_1_, pH_u_, DL, L*, a*, b*, WBSF
	2	10	10	HO	Conv.	
Channon et al. [35]	1	8	12	HO	Conv.	pH_1_, pH_u_, DL, L*, WBSF
	2	8	12	HO	Conv.	
	3	8	12	HO	Conv.	
	4	8	12	HO	Conv.	
	5	8	12	HO	Conv.	
	6	8	12	HO	Conv.	
	7	8	12	HB	Conv.	
	8	8	12	HB	Conv.	
	9	8	12	HB	Conv.	
	10	8	12	HB	Conv.	
Hambrecht et al. [18]	1	364	371	HO	Fast	pH_1_, pH_u_, DL, L*, a*, b*
	2	356	371	HO	Conv.	
Rees et al. [17]		12	12	HB	Conv.	pH_1_, pH_u_, DL, L*, a*, b*, WBSF
Hambrecht et al. [36]	1	47	49	HBO		pH_1_, pH_u_, DL, L*, a*, b*
	2	45	48	HBO		
Bertoloni et al. [19]	1	93	93	HO	Conv.	pH_u_, DL, L*, a*, b*
	2	76	76	HO	Conv.	
	3	49	49	HO	Conv.	
Lindahl et al. [37]	1	19	23	HO	Conv.	L*, a*, b*
	2	21	18	HO	Conv.	
	3	25	33	HO	Conv.	
	4	23	19	HO	Conv.	
Lammens et al. [38]	1	67	97	HB	Conv.	pH_1_, pH_u_
	2	91	97	HB	Fast	
	3	99	97	HB	Conv.	
	4	100	97	HB	Fast	
Van Heugten et al. [20]	1	18	10	HB	Conv.	pH_u_, DL, L*, a*, b*
	2	18	10	HB		
Shackelford et al. [39]		200	197		Fast	pH_u_, DL, L*, a*, b*, WBSF
Marcon et al. [15]		86	86	HB	Conv.	pH_1_, pH_u_, DL, L*, a*, b*, WBSF
Terlouw et al. [40]		10	10	HO	Conv.	pH_1_, pH_u_, DL, L*

Exp., experiment; ES type, type of electrical stunning; HO, head-only; HB, head-to-back; HBO, head-to-body; pH_1_, initial pH; pHu, ultimate pH, DL, drip loss; L*, lightness; a*, redness; b*, yellowness; WBSF, Warner–Bratzler shear force; Conv., conventional chilling at 1–4 °C; Fast chilling at temperatures from −10 °C to −35 °C in prechilling phase.

**Table 2 animals-12-01811-t002:** Descriptive statistics of parameters included in the meta-analysis.

Trait	Electrical Stunning						CO_2_ Stunning					
	*n*	Mean	SD	Min.	Max.	Median	*n*	Mean	SD	Min.	Max.	Median
pH_1_	1489	6.34	0.03	5.89	6.76	6.40	1582	6.43	0.04	6.05	6.89	6.39
pH_u_	5510	5.60	0.04	5.39	5.90	5.61	6229	5.63	0.05	5.37	5.86	5.63
DL (%)	1364	5.03	0.41	0.78	10.14	4.87	1423	3.71	0.43	0.64	8.86	4.28
L*	2780	50.47	0.97	41.70	60.00	50.47	3014	49.24	0.95	41.46	55.88	49.73
a*	2564	8.60	0.18	0.33	17.90	8.30	2768	7.73	0.20	0.28	17.00	7.70
b*	2564	6.27	0.17	2.70	19.80	5.81	2082	6.03	0.19	2.30	20.10	6.00
WBSF (kg)	398	6.89	0.69	3.53	18.20	5.80	427	6.66	0.71	3.32	18.80	5.59

*n*, number of animals; SD, standard deviation; pH_1_, pH measured between 45 min and 60 min after the slaughter; pH_u_, ultimate pH, DL, drip loss; L*, lightness; a*, redness; b*, yellowness; WBSF, Warner–Bratzler shear force.

**Table 3 animals-12-01811-t003:** Summary of the effect sizes of the stunning (electrical vs. CO_2_) on analysed pork quality traits in random-effect meta-analysis.

Trait	Group/ Subgroup	*n*	Effect Size	SE	95%CI	*p*-Value	I^2^
pH_1_	Overall	3071	−0.08	0.03	−0.151; −0.014	0.018	87.33%
	HB	1017	−0.14	0.04	−0.226; −0.057	<0.001	76.60%
	HBO	1639	0.01	0.08	−0.144; 0.167	0.881	95.05%
	HO	395	−0.04	0.03	−0.111; 0.021	0.185	16.37%
pH_u_	Overall	11,739	−0.01	0.02	−0.047; 0.025	0.550	93.50%
	HB	1073	−0.05	0.02	−0.092; −0.004	0.029	62.04%
	HBO	7208	0.03	0.04	−0.036; 0.103	0.360	97.61%
	HO	3061	−0.01	0.03	−0.071; 0.049	0.710	90.09%
	Conv.	4361	−0.01	0.02	−0.042; 0.023	0.570	71.77%
	Fast	7189	−0.07	0.04	−0,074; 0.075	0.980	97.77%
DL (%)	Overall	2787	0.68	0.19	0.312; 1.042	<0.001	73.49%
	HB	272	1.82	0.52	0.810; 2.834	<0.001	41.42%
	HBO	1579	0.01	0.268	−0.513; 0.537	0.95	77.65%
	HO	539	0.89	0.31	0.293; 1.495	<0.001	69.65%
	Conv.	1513	0.94	0.25	0.457; 1.427	<0.001	73.19%
	Fast	1081	−0.13	0.27	−0.659; 0.399	0.630	80.34%
L*	Overall	5794	1.29	0.41	0.482; 2.100	<0.001	91.56%
	HB	328	1.64	0.37	0.926; 2.366	<0.001	0%
	HBO	4203	1.29	0.87	−0.415; 3.001	0.140	97.24%
	HO	866	1.08	0.66	−0.211; 2.371	0.100	89.69%
	Conv.	4158	1.10	0.48	0.155; 2.040	0.020	90.59%
	Fast	1447	2.40	1.29	−0.125; 4.939	0.060	96.46%
a*	Overall	5360	0.80	0.07	0.656; 0.947	<0.001	99.86%
	HB	252	1.28	0.67	−0.035; 2.594	0.060	99.63%
	HBO	4056	0.83	0.12	0.589; 1.072	<0.001	99.95%
	HO	655	0.69	0.21	0.284; 1.090	<0.001	99.96%
	Conv.	3730	0.58	0.10	0.379; 0.783	<0.001	99.88%
	Fast	1447	0.88	0.16	0.569; 1.206	<0.001	99.57%
b*	Overall	5366	0.22	0.17	−0.117; 0.565	0.198	94.68%
	HB	252	0.42	0.23	−0.034; 0.883	0.070	66.79%
	HBO	4060	0.11	0.16	0.209; 0.434	0.490	91.12%
	HO	557	0.24	0.53	−0.804; 1.277	0.660	97.13%
	Conv.	3730	0.14	0.26	−0.307; 0.647	0.590	96.11%
	Fast	1447	0.37	0.25	−0.126; 0.861	0.140	86.05%
WBSF (kg)	Overall	825	0.04	0.196	−0.344; 0.429	0.820	0%

*n*, number of animals; HO, head-only electrical stunning; HB, head-to-back electrical stunning; HBO, head-to-body electrical stunning; Conv., conventional chilling at 1–4 °C; Fast chilling in temperatures from −10 °C to −35 °C in prechilling phase; pH_1_, measured between 45 min and 60 min after the slaughter; pH_u_, ultimate pH, DL, drip loss; L*, lightness; a*, redness; b*, yellowness; WBSF, Warner–Bratzler shear force; SE, standard error; 95% CI, confidence interval; I^2^ percentile of total variation due to heterogeneity.

## Data Availability

Data are available on request to the corresponding author.

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
