# Peer review of "Quantification of the Effects of Electrical and CO2 Stunning on Selected Quality Attributes of Fresh Pork: A Meta-Analysis"

_animals, 2022, doi:10.3390/ani12141811_

Round 1

Reviewer 1 Report

See uploaded file

Author Response

Dear Reviewer,

Thank you for taking the time to review our article. Thank you for your all constructive comments,  showing the detailed direction of necessary changes to a manuscript. Your comments are helpful not only in redrafting of this article but also in my research work.

Grateful author

Below are my responses to your comments:

Point 1: In general: The title, objective and conclusions are not in agreement. A definition of meat quality is lacking. A good, worse or bad quality and the normal (including variability) are not presented. Moreover, only 2 methods are included.

Response 1: You have a right, there is some inconsistency between the title and objective of the study. Firstly because title only in general refers to “meat quality”. Of course, meat quality is described according to different quality categories (as sensory/eating, technological, nutritional or ethical quality) consisting of various attributes. In this research, all quality parameters are associated with eating and technological quality, although they also refer to fresh/raw meat intended to consumption. According to e.g. Joo et al. 2013 “…the quality of fresh meat indicates its usefulness to the consumer and its acceptability for cooking. The important quality traits for fresh meat are (e.g.) colour, drip loss WHC,… tenderness”. Thus, on such kind of quality focused this meta-analysis. The overall definition of fresh pork quality and quality attributes important for fresh meat have been presented in lines 57-69, but with not precisive title and objective, I assume that such description may be misleading for a reader.

  The meta-analysis estimates the effects of interventions/treatment in comparison to e.g. control (without interventions/treatment) or another application showing the trends (they magnitude, direction, dispersion) for outcomes or how outcome can change with application of one treatment in comparison to other. In this approach, with quantitative outcomes, is hard clearly specify good, worse or bad quality however meta-analysis quantifies in which way, and how much (described by the effect size and 95%CI) the application of the treatment can change the value of the parameter in comparison to other treatment. That also is reason why only these of two stunning method were meta-analyzed. I know that these methods are not the only. Many researchers investigated other stunning methods with microwaves or different gases as an alternative to CO2, (argon, nitrogen, helium as individual factors or in the mixtures with CO2) or single-pulse ultra-high current stunning. However, most of them involved e.g. the welfare of animals, behavioural and physiological reactions or they stress responses to gases, or were conducted on other species. Only few , referred to selected meat quality traits or compared different variants of the same method (e.g. different gaseous mixtures, different CO2 concentrations – for example Nowak et al. 2007 or Atkinson et al. 2020, different parameters of ES e.g. May et al. 2022) on selected meat quality traits. I know only one research (Terlouw et al 2021) where the effects of ES in comparison to CO2 and N2O/CO2 mixture were addressed to selected meat quality traits.        

With regard to your constructive suggestions, I changed the title, clarified the objective, corrected the number of the third table (it was type error), and redrafted different parts of manuscript.

Summary

Point 2: 13 Or have effects.

Response 2: This has been corrected

Point 3: 15 The title describes just impact.

Response 3:This has been redrafted - lines 16-18

Point 4: 19 Beneficial, in which way? May have positive effects?

Response 4: Have negative effect but it has been changed lines - 27-29

Point 5: 21 Whom knows the future.

Response 5: This part of the sentence refers the interpretation of prediction interval (after Higgins et al.2009). However, this part of manuscript has been removed.

Abstract

Point 6: 29 …. Decreased or was lower …..

Response 6: This has been corrected

Point 7: 35 Why less beneficial ….?

Response 7: Due to negative effect of ES on analysed traits in overall lower pH1, greater DL higher L* in comparison to CO2 but However, this part of manuscript has been redrafted

Introduction

Point 8: 43 Should be unconscious and insensitive ….

Response 8: This has been corrected

Point 9: 57-69 There are no definitions here. In which way do you qualify the meat. Which quality is good, worse or bad or beneficial?

Response 9: As I wrote in preface, this study focused on several parameters that are important for fresh meat. Thus, in such way, the definition of fresh pork quality there was described.  

Point 10: 87-88 The objective is the effect of 2 stunning methods on selected parameters. There is a variability in ES and chilling.

Response 10: The objective has been redrafted as follows:

“the objective of this study was to apply a meta-analysis to quantify the effects of two stunning methods, electrical and gas with high concentration of carbon dioxide on selected quality attributes of fresh pork”

Results: Please, present your results and not your interpretation.

Point 11: 189 Why “negatively”. The difference is not significant.

Response 11: The overall differences were significant for those traits – Tab 3 , but I changed it. Line 202

Point 12: 200-202 This is a discussion.

Response 12: This has been removed from the section 

Point 13: 206 The pH was lower.

Response 13: This has been corrected in the text. Line 226

Point 14: 209 The drip loss was higher?

Response 14: Yes, DL was higher but it I redrafted the text - lines 231-232

Point 15: 212 Why “beneficial”.

Response 15: Yes you have a right, “with no effect” is more correct interpretation, or according to e.g. van Aert et al. 2019 shows on overestimation of the effect size, I redrafted the sentence lines 234-235.

Point 16: 216 Why negative?

Response 16: In this case, it refers to the sign (-) at a value of the effect size but , I redrafted the sentence line 244

Point 17: 221 …. higher …

Response 17: This has been corrected

Point 18: 223-224 This is interpretation.

Response 18: This has been removed from the section 

Discussion: Is too long and should be concentrated on discussion of results of parameters analysed.

Point 19: 253 What is normal meat?

Response 19: A meat without quality defects, but I replaced it by “Typically”

Point 20: 254 Which muscles: LD or SM? See Table 2 and not significant.

Response 20: This concern LD muscle and has been added into the text, description of results was 4 linen lower, in table (correctly) 3 the effect of stunning on pH1 is significant (p=0.018)

Point 21:289 See Table 2? In which way negative?

Response 21: HB stunned pigs had lower pH1 in comparison with CO2 although I redrafted the text – lines 327-329

Point 22:297 Table 2?

Response 22: This part concern table (correctly) 3 but I redrafted the text in present version line 359

Point 23:319 What is the consumer willing to purchase?

Response 23: It is behavioural intention of a consumer to buy a particular product, consumers do not want to by meat with high (visible) drip, ok fresh/raw meat is often packed in MAP with water absorbers, but not all meat in retailing is packed,  however this sentence has been removed

Point 24:329 Where is it demonstrated?

Response 24: This part concern table (correctly) 3 the text has been redrafted  in present version line 394-396

Point 25:350 Table 2?

Response 25: This part concern table (correctly) 3 the text has been redrafted  in present version line 419

Point 26:380 Why negatively?

Response 26: The application of ES increased L* but you have a right it is inappropriate or awkward word it has been redrafted  in present version line 439

Point 27:394 Whom tells the favour? Redness of meat can be very different and favoured.

Response 27: You have a right it is inappropriate and has been redrafted  in present version line 479

Point 28:396 Where to find?

Response 28: This part concern table (correctly) 3 the text has been redrafted  in present version line 479

Point 29:405 Why negatively?

Response 29: Has a negative effect because increases WBSF, but I changed it on “decreasing”

Point 30: Conclusions: should be reconsidered.

Response 30: This section has been redrafted 

Reviewer 2 Report

Comments on Manuscript:  animals-1770506 stunning

1.     To be frank, I though most of the results the meta-analysis showed had already been known.  This meta-analysis does not add that much to the literature and already present knowledge regarding the differences in electrical and CO2 stunning. Perhaps there is some new knowledge with the types of electrical stunning, but not much new information.
2.    The influences of stunning on metabolic processes shown in the paper provide a good review of how things like stunning influence pH and water holding capacity and would be appropriate for use in a graduate level class on meat science.
3.    The statistical analysis of the meta-analysis seems appropriate for this test.  It recognizes the heterogeneity between studies and attempted to assess publication bias as well.  These errors seem to be accounted for in the analysis.  The power of a meta-analysis is in the number of samples used.  Many times studies use a smaller number of animals for testing so when one takes many of these smaller tests, he can then conduct the meta-analysis to obtain results based upon larger numbers of animals.  The downside is that not each study is conducted in the same manner with the same types of animals so heterogeneity (error) creeps into the analysis.  So, it is a balance between having a sufficient number of animals and increasing errors that can influence the results.  This can be somewhat alleviated by carefully choosing the studies that are to be included in the work.  These authors seem to have been aware of these sorts of things and did screen the studies.
4.    Table 2 does show relatively high heterogeneity values so it is good these have been accounted for in the analysis; however, I do not know how this influences the acceptability of the results.  That is to say, how much do I trust a result if it has a high heterogeneity value?  Does that mean that there is high uncertainty about the result?  I think general practice shows the same results as the paper shows in trends, but I am not sure that the absolute results can be used.
5.    The presentation of the data in the two tables seems appropriate for this kind of analysis.
6.    Note that this study is completely based upon a computer analysis of the data.  It is not validated in the fact that the results have not been tested against other studies.  This is typical with meta-analyses.  Perhaps the next step would be to do so.
7.    Since the study did not use any animals, no ethics statement was required.

This is an interesting paper that combines and summarizes the influences of stunning by different means on meat quality.  The main issue, though, is regarding the English used in the paper.  There are many awkwardly worded sentences.  The use of CO2 rather than CO2 was present (which I know is hard to always get right).  I think a review by a good English editor would fix the problems observed.

Author Response

Dear Reviewer,

Thank you for taking the time to review our article. Thank you for your all constructive comments,  showing the detailed direction of necessary changes to a manuscript. Your comments are helpful not only in redrafting of this article but also in my research work.

Grateful author

Point 1: 1.     To be frank, I though most of the results the meta-analysis showed had already been known.  This meta-analysis does not add that much to the literature and already present knowledge regarding the differences in electrical and CO2 stunning. Perhaps there is some new knowledge with the types of electrical stunning, but not much new information.

Response 1: To my knowledge, there is no published meta-analysis on the effect of stunning methods on meat quality traits. This meta-analysis covers all available papers that pass screening process. Currently, electric stunning and exposure to carbon dioxide are the most frequently used methods for pigs stunning. I know that these methods are not the only. Many researchers investigated other stunning methods with microwaves or different gases as an alternative to CO2, (argon, nitrogen, helium as individual factors or in the mixtures with CO2) or single-pulse ultra-high current stunning. I know that will be better to compare in sub-groups ES with different gas methods. But, the main issue is a topic and experimental design of studies investigating effects of various stunning methods. The most of them, even though they were conducted on pigs, referred to involved e.g. the welfare of animals, behavioural and physiological reactions or they stress responses to gases. Only few, referred to pigs and selected  pork quality traits e.g. Lionch et al 2012 or compared different variants of the same method (e.g. different gaseous mixtures, different CO2 concentrations – for example Nowak et al. 2007 or Atkinson et al. 2020, different parameters of ES e.g. May et al. 2022) on selected meat quality traits. I know only one research (Terlouw et al 2021) where the effects of ES in comparison to CO2 and N2O/CO2 mixture were addressed to selected meat quality traits. Thus, inclusion of other methods into the meta-analysis was limited.

Point 2: 2.    The influences of stunning on metabolic processes shown in the paper provide a good review of how things like stunning influence pH and water holding capacity and would be appropriate for use in a graduate level class on meat science.

Response 2: Thank you for your opinion,

Point 3: 3.    The statistical analysis of the meta-analysis seems appropriate for this test.  It recognizes the heterogeneity between studies and attempted to assess publication bias as well.  These errors seem to be accounted for in the analysis.  The power of a meta-analysis is in the number of samples used.  Many times studies use a smaller number of animals for testing so when one takes many of these smaller tests, he can then conduct the meta-analysis to obtain results based upon larger numbers of animals.  The downside is that not each study is conducted in the same manner with the same types of animals so heterogeneity (error) creeps into the analysis.  So, it is a balance between having a sufficient number of animals and increasing errors that can influence the results.  This can be somewhat alleviated by carefully choosing the studies that are to be included in the work.  These authors seem to have been aware of these sorts of things and did screen the studies.

Response 3: Thank you for your opinion, I try to prepare this meta-analysis including all methodical steps as screening process (choosing data from the pigs, measured in the same muscle, time and used the same measure method) and calculations (I2, publication bias and PI).  

Point 4: 4.    Table 2 does show relatively high heterogeneity values so it is good these have been accounted for in the analysis; however, I do not know how this influences the acceptability of the results.  That is to say, how much do I trust a result if it has a high heterogeneity value?  Does that mean that there is high uncertainty about the result?  I think general practice shows the same results as the paper shows in trends, but I am not sure that the absolute results can be used.

Response 4: Indeed, the presence of heterogeneity across studies is a major concern in meta-analysis. Firstly we should use of random-effect model that assumes that there is heterogeneity. In such way, the effect size and 95%CI ( is wider than in fixed model because takes account uncertainty) are closer to means of individual studies, but may be overestimated. Of course, by the use of sub-groups, I try to find a variable or variables which explains this heterogeneity. However, only few had low heterogeneity or were homogenous. We could allow for the heterogeneity in our analysis with wider 95CI and random-effects model, but additionally according to Higgins et al 2009 I reported prediction interval which should be reported in presence of heterogeneity and is interpret as a description of the range of observed effect sizes. I also changed conclusions addressing the results to PI.

Point 5: 5.    The presentation of the data in the two tables seems appropriate for this kind of analysis.

Response 5: Thank you for your opinion

Point 6: 6.    Note that this study is completely based upon a computer analysis of the data.  It is not validated in the fact that the results have not been tested against other studies.  This is typical with meta-analyses.  Perhaps the next step would be to do so.

Response 6: I agree, I thought about such approach but it concern glycogen changes in muscles but it is a work on future

  1. Since the study did not use any animals, no ethics statement was required.

Point 7: This is an interesting paper that combines and summarizes the influences of stunning by different means on meat quality.  The main issue, though, is regarding the English used in the paper.  There are many awkwardly worded sentences.  The use of CO2 rather than CO2 was present (which I know is hard to always get right).  I think a review by a good English editor would fix the problems observed.

Response 7: I redrafted the manuscript according to suggestions of one of reviewers, I also did everything to improve the language of manuscript.

Reviewer 3 Report

This is a meta-analysis work on the effect of stunning on pork quality. The study does not provide new knowledge or methods that have not been referenced in the studies used to develop the analysis, mainly assuming that most of the studies used in the meta-analysis are prior to 2004, with the exception of 2 works published in 2019 and 221 without large expression in the data set. It seems redundant to conclude that further studies are needed to understand the effect of stunning on pork to better understand the effect of stunning on pork and to increase the reliability of the results in potential future meta-analyses. analysis. In the end, what else are more studies needed to produce more meta-analyses!!!!

Author Response

Dear Reviewer,

Thank you for taking the time to review our article. Thank you for your all constructive comments,  showing the detailed direction of necessary changes to a manuscript. Your comments are helpful not only in redrafting of this article but also in my research work.

Grateful author

Point 1: This is a meta-analysis work on the effect of stunning on pork quality. The study does not provide new knowledge or methods that have not been referenced in the studies used to develop the analysis, mainly assuming that most of the studies used in the meta-analysis are prior to 2004, with the exception of 2 works published in 2019 and 221 without large expression in the data set. It seems redundant to conclude that further studies are needed to understand the effect of stunning on pork to better understand the effect of stunning on pork and to increase the reliability of the results in potential future meta-analyses. analysis. In the end, what else are more studies needed to produce more meta-analyses!!!!

Response 1: To my knowledge, there is no published meta-analysis on the effect of stunning methods on meat quality traits. This meta-analysis covers all available papers that met all screening criteria. Electric stunning and high CO2 are the most frequently used for stunning of pigs. Exactly most of them is to 2004, but other are from later years. I know that these methods are not the only. Many researchers investigated other stunning methods with microwaves or different gases as an alternative to CO2, (argon, nitrogen, helium as individual factors or in the mixtures with CO2) or single-pulse ultra-high current stunning. I know that will be better to compare in sub-groups ES with different gas methods. But, the main issue is a topic and experimental design of studies investigating effects of various stunning methods. The most of them, even though they were conducted on pigs, referred to involved e.g. the welfare of animals, behavioural and physiological reactions or they stress responses to gases. Only few, referred to pigs and selected  pork quality traits e.g. Lionch et al 2012 or compared different variants of the same method (e.g. different gaseous mixtures, different CO2 concentrations – for example Nowak et al. 2007 or Atkinson et al. 2020, different parameters of ES e.g. May et al. 2022) on selected meat quality traits. I know only one research (Terlouw et al 2021) where the effects of ES in comparison to CO2 and N2O/CO2 mixture were addressed to selected meat quality traits. Thus, inclusion of other methods into the meta-analysis was limited. Exactly, conclusion “that further studies are needed to understand the effect of stunning on pork to better understand the effect…” was improper and was removed. I really apologies for all improper phrases. I redrafted the title, different parts of manuscript and conclusions.

Round 2

Reviewer 1 Report

In general

The text is improved acc0ording the comments.

Abstract

DL, HB, HO and HBO Need explanation.

There are several writing errors, please, control text. 

Author Response

Dear Reviewer,

Thank you again for taking the time to review our article. Thank you for your all constructive comments, showing the detailed direction of necessary changes to a manuscript. Your comments are helpful not only in redrafting of this article but also in my research work.

Grateful author

Point 1: In general

The text is improved acc0ording the comments.

Abstract

DL, HB, HO and HBO Need explanation.

There are several writing errors, please, control text. 

Response 1:  The all abbreviations (DL, HB, HO and HBO) has been explained in abstract, I also carefully checked the text of the manuscript and corrected all noticed writing errors.

With kind regards

Author

Reviewer 3 Report

 apologize for my insistence but the study does not bring anything new and is nothing more than a reinterpretation of results from previous works, but without specifying the experimental conditions in which each of them took place, so the discussion turns out to be somewhat speculative.

Author Response

Point 1: apologize for my insistence but the study does not bring anything new and is nothing more than a reinterpretation of results from previous works, but without specifying the experimental conditions in which each of them took place, so the discussion turns out to be somewhat speculative.

Response 1: Dear Reviewer, thank you for your opinion regarding the results of this work. The meta-analysis as the method is not ideal tool. In the scientific community has been discussion concerning pros and cons of meta-analyses. However, in comparison to narrative or systematic reviews, is a powerful tool to cumulate and summarize the knowledge in a research field through statistical instruments, and to identify the overall measure of a treatment’s effect by combining several individual results. This meta-analysis have been performed with all rules and standards recommended for such procedure. An important role in meta-analytic approach have screening process and inclusion criteria. Broad inclusion criteria could increase the variability (error) between studies, narrow inclusion criteria decreases variability between studies but is harder to find appropriate and more recent studies regarding the topic of the study, with relevant outcomes and they variability measure in each study. Despite of screening process (in this study data from the same specie, measured in the same muscle, time and used the same measure method) and calculations (using random-effects model), in random-effects sub-group analysis (with all available to extract from publications and common for experiments factors as moderators) was tested whether the pooled effect sizes found in these subgroups differ significantly from each other. The random-effects model assumes differences between studies and experiments and vary, in comparison to fixed-effects model, both in threshold and width of confidential interval (wider in comparison to CI from fixed-effects model) and significance of results.

In some cases, meta-analysis (analysing different studies trends) may show or lead to support an overall, derived from several individual studies, knowledge concerning the effect of application (its trend or direction) on some traits, but in comparison to results or inferences from individual studies “it still adds value to knowledge by quantification assessment of the effect” – that cited opinion is not my own, but was taken from one of the many discussion concerning pros and cons meta-analyses. I respect you opinion but meta-analysis specifying the experimental conditions in which each, included into computing experiment, took place, in my opinion is not possible to perform. But, as I mentioned earlier, such role, in some extent, has random-effects sub-group analysis (with all available to extract from publications and common for experiments factors as moderators. There was available to extract as moderators type of ES and chilling method. I know that for example includeing the type of restrainer as moderator could also show different, maybe more robust results but in general it was not specify in publications. Computed results showed that for DL and L* there was no differences between two ES methods and CO2 when fast chilling was applied. In contrast, the inferences from individual studies didn’t provide such results (recommendations) for meat processors.

Aditionally, as I mentioned in previous response, there is no published meta-analysis on the effect of stunning methods on meat quality traits. In 2020, a group of French scientists ( Prache, S., Santé-Lhoutellie, V., Adamiec, C., Astruc, T., Baéza-Campone, E., Bouillot, P.E., Clinquart, A., Feidt, C., Fourat, E., Gautron, J., Guillier, L., Kesse-Guyot, E., Lebret, B., Lefèvre, F., Martin, B., Mirade, P.S., Pierre, F., Raulet, M., Rémond, D., Sans, P., Souchon, I., Donnars, C.) published report “The quality of animal-based food related to animal production and processing conditions”. The objective of this study was to characterise the quality of foods products according to animal production and processing conditions, focusing on the determinants of the properties that constitute quality. In this report the authors highlighted issues related to production, processing and consumption of animal-based foods, and summarised report with the list of the main options for publication and research needs. As a lessons from results they highlighted “lack of meta-analyses on the quantification of the effects of production and processing factors on food quality”. As research needs they showed to ”Carry out meta-analyses to obtain a more robust, quantitative assessment of the effect of different factors affecting quality”.

With kind regards

Author